

# Dynamic stacking ensemble for cross-language code smell detection

Hamoud Aljamaan[1,2]

[1] Information and Computer Science Department, King Fahd University of Petroleum and Minerals, Dhahran, Saudi Arabia
[2] Interdisciplinary Research Center for Finance and Digital Economy, King Fahd University of Petroleum and Minerals, Dhahran, Saudi Arabia

## ABSTRACT

Code smells refer to poor design and implementation choices by software engineers that might affect the overall software quality. Code smells detection using machine learning models has become a popular area to build effective models that are capable of detecting different code smells in multiple programming languages. However, the process of building of such effective models has not reached a state of stability, and most of the existing research focuses on Java code smells detection. The main objective of this article is to propose dynamic ensembles using two strategies, namely greedy search and backward elimination, which are capable of accurately detecting code smells in two programming languages (*i.e.*, Java and Python), and which are less complex than full stacking ensembles. The detection performance of dynamic ensembles were investigated within the context of four Java and two Python code smells. The greedy search and backward elimination strategies yielded different base models lists to build dynamic ensembles. In comparison to full stacking ensembles, dynamic ensembles yielded less complex models when they were used to detect most of the investigated Java and Python code smells, with the backward elimination strategy resulting in less complex models. Dynamic ensembles were able to perform comparably against full stacking ensembles with no significant detection loss. This article concludes that dynamic stacking ensembles were able to facilitate the effective and stable detection performance of Java and Python code smells over all base models and with less complexity than full stacking ensembles.

## INTRODUCTION

Code smells refer to poor design and implementation choices by software engineers that compromise overall software quality (*Yamashita & Moonen, 2013*). During the software development process, there are many types of code smells that might be injected into the code, such as the following: duplicated code, large class, long method, *etc.* In response, software engineers are always encouraged to perform routine software refactoring to counter these code smells and thereby improve the overall software maintainability and quality (*Fowler, 1999*). However, identifying these refactoring opportunities (*i.e.*, code smells) is not an easy and straightforward task, and code smell identification is the first

Corresponding author
Hamoud Aljamaan,
hjamaan@kfupm.edu.sa

crucial step in a successful software refactoring process (*Kim, Zimmermann & Nagappan, 2012*).

Several approaches have been proposed to detect code smells in source code, including metrics-based (*Charalampidou, Ampatzoglou & Avgeriou, 2015*), rule-based (*Moha et al., 2009*), machine learning-based (*Fontana et al., 2013*), and deep learning-based approaches (*Zhang & Jia, 2022*). Ensemble learning (*Rokach, 2010*) has proven to be a reliable machine learning paradigm, wherein multiple base models prediction outputs are aggregated into a final model output. Nevertheless, the majority of the empirical studies conducted in this area have focused on the utilization of conventional machine learning models (*Di Nucci et al., 2018*; *Arcelli Fontana et al., 2016*; *Alazba & Aljamaan, 2021*; *Aljamaan, 2021*) within the limited context of Java code smells. Code smell detection using ensemble learning has not yet reached a stable state (*Di Nucci et al., 2018*; *Azeem et al., 2019*; *Al-Shaaby, Aljamaan & Alshayeb, 2020*), and code smell detection studies should include smells from languages other than Java to investigate the generalizability of machine learning models in code smells detection.

In this article, we propose to build dynamic stacking ensembles using two strategies: greedy search and backward elimination. Our main hypothesis is that dynamic ensembles will produce less complex stacking ensembles with stable and superior code smells detection performance across multiple languages. We will empirically evaluate the effectiveness of dynamic stacking ensembles in code smell detection across four Java code smells and two Python code smells. Dynamic ensembles detection performance will be compared with all base models; we will also assess whether these ensembles will yield in less complex models in comparison to full stacking ensembles without significant performance loss in Java and Python code smells detection. Our article contributions can be summarized as follows:

- Cross-language code smell detection. We extend the generalizability of our approach in code smell detection across different programming languages.
- Reduced complexity. Proposed approach will simplify the stacking ensemble structure, making it less computationally expensive without compromising the detection performance.
- Comprehensive empirical evaluation. We validate our approach using a diverse Java and Python datasets with multiple code smell types, providing a thorough assessment of the proposed approach robustness and effectiveness.

This article is structured as follows: "Literature Review" discusses the literature review with the targeted research gaps. "Dynamic Stacking Ensemble" explains the dynamic ensembles building approaches. "Research Methodology" states our research objective and followed methodology. "Empirical Study Design" outlines the empirical study design details. "Results and Discussions" presents the empirical study results and answers the research questions. "Threats to Validity" discusses the empirical study identified threats to validity and the measures taken to address them. "Research Implications" discusses the

implications of our findings. "Conclusion" concludes the article with future work directions.

## LITERATURE REVIEW

In this section, we provide an overview of the related existing studies. Specifically, we survey the literature on using ensemble learning for code smell detection. Studies included in the survey must have a rigorous empirical design and published in reputable journals and conferences within the last five years. This review not only contextualizes our current study, but also identifies gaps and opportunities for further exploration, setting the stage for the novel insights and contributions this article aims to deliver in the context of code smell detection.

In early research, researchers used metrics (*Charalampidou, Ampatzoglou & Avgeriou, 2015*) and rule-based (*Moha et al., 2009*) approaches in code smell detection. In metrics-based approach, software engineers needed to set the right thresholds for each metric to detect targeted code smell, while in rule-based approaches, rules are manually created by domain experts to detect each code smell type. Both approaches are challenging tasks. Therefore, recently machine learning models have been utilized in code smell detection by extracting code metrics to build machine learning models and create detection rules and select thresholds automatically.

Code smell detection using machine learning models has been an active research area recently. Researchers have tackled the code smell detection problem using various approaches, such as treating it as a binary classification problem (*Mahalakshmi et al., 2023*), addressing smell severity as a multi-class classification problem (*Rao et al., 2023*), or handling it as a multi-label classification problem (*Yadav, Rao & Mishra, 2024*). In our article, we aim to detect code smells as a binary classification problem, concentrating on distinguishing between the presence and absence of code smells at both the class and method levels.

The majority of the conducted empirical studies in this area focused on the utilization of conventional machine learning models, such as, support vector machines (*Di Nucci et al., 2018; Arcelli Fontana et al., 2016; Alazba & Aljamaan, 2021; Aljamaan, 2021*), Logistic Regression (*Di Nucci et al., 2018; Arcelli Fontana et al., 2016; Alazba & Aljamaan, 2021; Aljamaan, 2021*), *etc*. *Arcelli Fontana et al. (2016)* conducted the largest experiment on code smell detection and provided a benchmark dataset of Java code smells. They showed that random forest ensemble approach outperforms single models, which highlights the effectiveness of using ensemble learning in code smell detection (*Arcelli Fontana et al., 2016; Alazba & Aljamaan, 2021*). Nevertheless, code smell detection using ensemble machine learning has not yet reached a stable state (*Di Nucci et al., 2018; Azeem et al., 2019; Al-Shaaby, Aljamaan & Alshayeb, 2020*).

*Arcelli Fontana et al. (2016)* implemented AdaBoost ensemble model, wherein many classifiers of the same type were combined. In the conducted experiment, it was observed that the implementation of the boosting ensemble model yielded varying effects on the performance of the basic classifier. In certain instances, the performance was enhanced, while in other instances, it was diminished. This study was replicated

by *Di Nucci et al. (2018)* with a focus on identifying and discussing the limitations observed in the original experiment. They employed the same experimental settings, but they augmented each code smell dataset with real-world instances. The findings of their study indicate that the accuracy of the performance was worse in comparison to the original study. Therefore, the authors recommended that future research should focus on refining the machine learning algorithms to improve their generalizability across different datasets.

*Alazba & Aljamaan (2021)* investigated the detection performance of 14 individual classifiers and three stacking heterogeneous ensemble models. Their findings demonstrate that the ensemble models showed significantly higher performance in detecting both class-level and method-level code smells compared to the individual classifiers. This suggests that combining the predictions of multiple classifiers can improve the overall detection performance. In addition, the study highlights the potential of using stacking ensembles as a reliable approach for code smell detection. In another study, *Aljamaan (2021)* conducted an empirical investigation of the voting ensemble strategy, including heterogeneous classifiers. In this approach, the prediction outputs generated by several classifiers were combined using soft voting. By leveraging the strengths of different classifiers, the voting ensemble approach can effectively detect a wide range of code smells. The statistical analysis of pairwise comparisons reveals that the voting ensemble achieved higher performance in detecting all code smells.

In a recent study, *Dewangan et al. (2022)* employed a selection of ensemble machine learning algorithms, including Adaboost, Bagging, Max voting, gradient boosting, and XGBoosting, as well as two deep learning algorithms, namely artificial neural network and convolutional neural network. After conducting a comparative analysis of the results obtained from the different approaches on different datasets, it was concluded that the Max voting algorithm has the highest overall accuracy. In another study, *Kaur & Kaur (2021)* utilized two ensemble approaches, namely Bagging and random forest, employing two aggregation techniques: Majority Voting and Smaller is Heavier (*Saeys, Abeel & Van de Peer, 2008*). Although the results are positive, the utilization of ensemble learning methodologies necessitates extensive validation across diverse datasets before standardizing the use of ensemble learning approaches.

*Zhang & Jia (2022)* introduced a novel methodology that integrates deep learning techniques with ensemble methods. The optimization of these two methods is achieved through three key aspects: incorporating the attention mechanism, modifying the model structure, and utilizing Snapshot ensemble approach (*Huang et al., 2017*). The fundamental concept of the Snapshot ensemble approach involves the periodic adjustment of the learning rate. After saving the M total snapshots, the integration process is performed using a subset of the last m snapshots (where m < M). The outcome is determined by calculating the weighted average of all the m snapshots. Overall, by combining these three aspects, the model achieves enhanced performance.

Recently, *Alazba, Aljamaan & Alshayeb (2024b)* proposed Code Representation with Transformers (CoRT), a new code smell detection approach utilizing self-supervised learning. CoRT was trained to learn the structural and semantic features that are useful for multiple downstream tasks, such as the detection of two class-level smells (God Class and

**Table 1 Ensemble learning models utilized in code smells detection.**

| Ref. | Dataset | Prog. Lang. | Code smell types* | | | | | | | | Input | Ensemble approach |
|------|---------|-------------|-----|-----|-----|-----|-----|-----|-----|-----|-------|-------------------|
| | | | DC | FE | GC | LC | LM | LPL | SS | Other | | |
| *Dewangan et al. (2022)* | *Arcelli Fontana et al. (2016)* | Java | ✓ | ✓ | ✓ | | ✓ | | | | Metrics | Adaboost, Bagging, Max voting, Gradient boosting, and Xgboosting. |
| *Alazba & Aljamaan (2021)* | *Arcelli Fontana et al. (2016)* | Java | ✓ | ✓ | ✓ | | ✓ | ✓ | ✓ | | Metrics | Static stacking ensemble |
| *Zhang & Jia (2022)* | *Liu et al. (2019)* | Java | | ✓ | | | | | | | Code | Snapshot ensemble. |
| *Aljamaan (2021)* | *Arcelli Fontana et al. (2016)* | Java | ✓ | ✓ | ✓ | | ✓ | ✓ | ✓ | | Metrics | Soft voting ensemble. |
| *Kaur & Kaur (2021)* | In-house | Java | ✓ | | ✓ | | | | | ✓ | Metrics | Bagging and Random Forest. |
| *Di Nucci et al. (2018)* | In-house | Java | ✓ | ✓ | ✓ | | ✓ | | | | Metrics | ADABOOSTM1 boosting. |
| *Arcelli Fontana et al. (2016)* | In-house | Java | ✓ | ✓ | ✓ | | ✓ | | | | Metrics | Boosting. |
| This work | *Arcelli Fontana et al. (2016), Sandouka & Aljamaan (2023)* | Java & Python | ✓ | ✓ | ✓ | ✓ | ✓ | ✓ | ✓ | | Metrics | Novel Dynamic Stacking ensemble. |

**Note:**
* DC, Data Class; FE, Feature Envy; GC, God Class; LC, Large Class; LM, Long Method; LPL, Long Parameter List; SS, Switch Statements; Other, Brain Method, Shotgun Surgery, Dispersed coupling, and Message Chains.

Data Class), and two method-level smells (Feature Envy and Long Method). Detection approach was assessed against supervised and feature-based approaches and CoRT achieved high detection performance in all code smells. Later, *Alazba, Aljamaan & Alshayeb (2024a)* introduced Model Representation with Transformers (MoRT), an automated detection approach of class diagram smells using self-supervised learning.

Overall, this survey highlights the potential of ensemble learning in improving code smell detection and suggests further research and experimentation with different ensemble techniques. Table 1 summarizes the related studies that utilized ensemble learning in code smells detection, showing the different datasets, code smells types, and ensemble approaches. Most of these studies have significant limitations, as follows:

- **Gap 1:** Generalizability. Majority of empirical studies in this area have focused on Java code smells to evaluate the performance of their proposed techniques. However, it would be beneficial to explore other programming languages as well, to ensure the generalizability, validity, and applicability of the findings.
- **Gap 2:** Ensemble models stability. Code smell detection using ensemble learning has not yet reached a stable state. The performance of these models can be inconsistent, and they often require extensive tuning and validation.
- **Gap 3:** Ensemble models complexity. Many ensemble learning models, such as full stacking ensembles, result in complex models that are computationally expensive and

difficult to interpret. There is a need for approaches that allow ensemble models to achieve high performance without such complexity.

In this research, we aim to address these gaps by introducing a dynamic stacking ensembles approach, which is designed to be less complex while maintaining a robust performance across different programming languages (Java and Python). This approach will provide a more generalizable, stable, and efficient detection method for code smell detection.

## DYNAMIC STACKING ENSEMBLE

Stacking ensemble (*Wolpert, 1992*) is a well known ensemble learning algorithm that consists of a two layered architecture, as illustrated in Fig. 1. In the first layer, the base models are trained using the training dataset, and then, the base models predictions are then aggregated to construct a new training dataset for the second layer. In the second layer, the new constructed dataset is used to train a meta learner to produce the final stacking ensemble prediction.

A stacking ensemble algorithm can be formalized as outlined in Table 2, into three main steps: train ensemble base models; construct new dataset; and train meta learner. Stacking ensembles are often heterogeneous in nature, meaning that the base models are from different machine learning classification families (*e.g.* decision trees, support vector machines). Base models are expected to have different skills on the training dataset that can benefit the meta learner training. For classification purposes, stacking ensemble meta learners are recommended to be logistic regression (*Alazba & Aljamaan, 2021*) due to the linearity nature of the base models predictions.

In this article, we propose an enhancement to Full Stacking Ensembles (FSE) by introducing Dynamic Stacking Ensembles (DSE) with the following two strategies related to stacking base models selection, namely: Greedy Search (GS) and Backward Elimination (BE). These two strategies will be used to select base models from a candidate base models list allowing the stacking ensemble to be created dynamically by selecting different base models in different datasets. Dynamic stacking complexity will be determined by the number of base models required to build the dynamic stacking ensemble. The lower number of base models indicates less complex dynamic stacking ensemble model.

### Greedy search

Greedy search (GS) (*Cormen et al., 2022*) is a heuristic algorithm that aims to find a global optimum solution by making locally optimal choices at each step. Dynamic Stacking Ensemble with Greedy Search algorithm (DSE-GS) is the first strategy to build a dynamic stacking ensemble by selecting the ensemble base models in a forward greedy selection approach, as outlined in Table 3. Our algorithm begins with an empty list of base learners, and assumes that a stacking ensemble can still be built using only base model. Thus, in the first initial step, we will iterate over all base models, and build stacking ensembles for each base model accordingly and select the base model with the highest performance. The algorithm will then iterate over the remaining base models and select one base model in

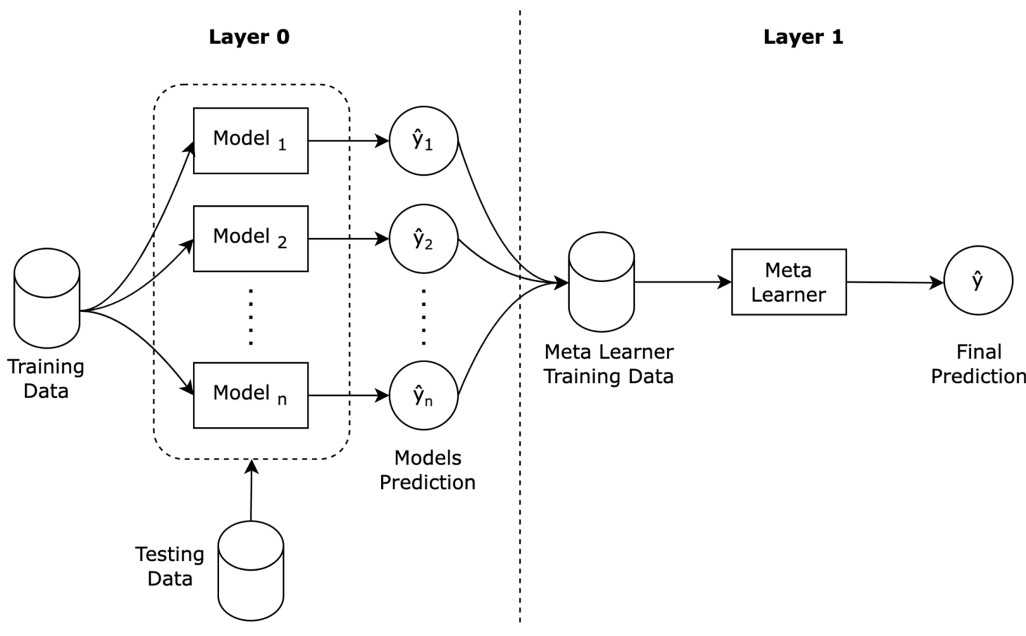

**Figure 1 Stacking ensemble architecture.**

---

**Table 2 Stacking ensemble algorithm.**

**Input:**

Training data $\mathscr{D} = \{x_i, y_i\}_{i=1...m}$, where $x_i$ is the feature vector of the $i$-th sample, $y_i$ is the target value, and $m$ is the number of samples.

Base Models $\mathscr{B}_{i=1...N}$: Set of $N$ base models.

***Step 1: Train Base Models***

**for** $n \leftarrow 1$ to $N$ **do**               ▷ $N$: number of base models

  Train base model $\mathscr{B}_n$ on $\mathscr{D}$           ▷ $\mathscr{B}_n$: $n$-th base model

**end for**

***Step 2: Construct New Dataset***

**for** $i \leftarrow 1$ to $m$ **do**

  Construct a new dataset $\mathscr{D}' = \{x'_i, y_i\}$,      ▷ $x'_i$: base model predictions vector

   where $x'_i = \{\mathscr{B}_1(x_i), \mathscr{B}_2(x_i), \dots, \mathscr{B}_N(x_i)\}$

**end for**

***Step 3: Train Meta Learner***

Train a meta learner $\mathscr{M}$ based on $\mathscr{D}'$

**Output:**

Stacking Ensemble $S(x) = \mathscr{M}(\mathscr{B}_1(x), \mathscr{B}_2(x), \dots, \mathscr{B}_N(x))$     ▷ $x$: input

---

each iteration with the greatest contribution to the stacking ensemble performance. Lastly, the algorithm will halt when none of the remaining base models will contribute to the stacking ensemble performance.

**Table 3 Greedy search algorithm.**

**Input:**

Training data $\mathscr{D} = \left\{\mathbf{x}_i, \mathrm{y}_i\right\}_{i=1\ldots m}$, where $\mathbf{x}_i$ is the feature vector of the $i$-th sample, $\mathrm{y}_i$ is the target value, and $m$ is the number of samples.

Base Models $\mathscr{B}_{i=1\ldots n}$: Set of $n$ base models.

***Step 1: Select First Base Model***

**for** $i \leftarrow 1$ to $n$ **do**            ▷ $n$: number of base models

     Build Stack $S$ with $\mathscr{B}_i$       ▷ $S$: stack built with the $i$-th base model

     **if** $skill(S)$ is highest:       ▷ $skill$: measures the detection performance

         $model \leftarrow \mathscr{B}_i$       ▷ $model$: selected base model

**end for**

$\mathscr{B}' \leftarrow model$       ▷ $\mathscr{B}'$: selected set of base models

$\mathscr{B} \leftarrow \mathscr{B} - model$       ▷ $\mathscr{B}$: original set of base models

***Step 2: Forward Base Model Selection***

**while** *a model has been selected* **do**

     **for** $i \leftarrow 1$ to $n - 1$ **do**

         Build Stack $S$ with $\mathscr{B}'$       ▷ $\mathscr{B}'$: set of selected base models

         Build Stack $S'$ with $\mathscr{B}' + \mathscr{B}_i$

         **if** $skill(S') > skill(S)$:       ▷ $skill$: measures the detection performance

            $model \leftarrow \mathscr{B}_i$       ▷ $model$: selected base model for inclusion

     **end for**

     **if** *a model has been selected*:

         $\mathscr{B}' \leftarrow \mathscr{B}' + model$       ▷ $\mathscr{B}'$: selected set of base models

         $\mathscr{B} \leftarrow \mathscr{B} - model$       ▷ $\mathscr{B}$: original set of base models

**end while**

**Output:**

New Base Models $\mathscr{B}'_{i=1\ldots s}$       ▷ $s$: set of selected base models

## Backward elimination

Backward elimination (BE) (*Guyon & Elisseeff, 2003*) is a recursive elimination algorithm that iteratively removes the least contributing models, thus, refining the models selection based on their contributions. Dynamic Stacking Ensemble with Backward Elimination algorithm (DSE-BE) is another strategy to select a refined list of base models to build a dynamic stacking ensemble. In BE algorithm as outlined in Table 4, we start with the full base models list to build the stacking ensemble, then, iterate over each base model to build a new stacking ensemble without the selected base model. Remove the base model from the list that yielded a positive impact on the stacking ensemble performance. The algorithm will be repeated until none of the base models removal will yield a positive impact on the stacking ensemble performance.

**Table 4 Backward elimination algorithm.**

**Input:**

Training data $\mathscr{D} = \{\mathbf{x}_i, y_i\}_{i=1\dots m}$, where $\mathbf{x}_i$ is the feature vector of the $i$-th sample, $y_i$ is the target value, and $m$ is the number of samples.

Base Models $\mathscr{B}_{i=1\dots n}$: Set of $n$ base models.

*Step: Backward Model Elimination*

**while** *a model has been selected* **do**

    **for** $i \leftarrow 1$ to $n-1$ **do**                         ▷ $n$: number of base models

        Build Stack $S$ with $\mathscr{B}$                 ▷ $S$: stack built with $\mathscr{B}$

        Build Stack $S'$ with $\mathscr{B} - \mathscr{B}_i$     ▷ $S'$: stack built excluding the $i$-th base model

        **if** $skill(S') \geq skill(S)$:        ▷ $skill$: measures the detection performance

            $model \leftarrow \mathscr{B}_i$            ▷ $model$: base model to be removed

    **end for**

    **if** *a model has been selected*:

        $\mathscr{B} \leftarrow \mathscr{B} - model$           ▷ $\mathscr{B}$: Updated set of base models

**end while**

**Output:**

Refined Base Models $\mathscr{B}_{i=1\dots s}$        ▷ $s$: set of selected base models

# DISCUSSION

In this article, we aim to employ both GS and BE strategies to optimize the selection of base models in our dynamic stacking ensemble approach. The choice of either approach can be determined by their respective strengths and limitations. GS is a heuristic algorithm that makes locally optimal choices at each step with the goal of finding a global optimum solution. Despite not guaranteeing a global optimal solution due to lack of backtracking, GS strategy is simple and fast in navigating large solution spaces, making it a practical solution to select the first initial base model to build the dynamic stacking ensemble. Meanwhile, BE strategy is a recursive base model elimination approach that starts with a full base models and iteratively removes the least significant base models. Despite being computationally intensive, BE allows for a thorough evaluation of each base model contribution to the dynamic stacking ensemble ensuring that the final base models list includes only those that positively contribute to ensemble performance.

In this research, we aim to investigate the applicability of constructing dynamic stacking ensembles using both strategies in cross-language code smell detection. Our hypothesis is that using both strategies, we will be able to construct less complex DSE in contrast to FSE, with comparable detection performance (*Zhou, 2012*; *Chatzimparmpas et al., 2020*). DSEs are designed to reduce the number of base models included in the final stacking ensemble, thus reducing the overall space and inference time complexity. The stacking space complexity is computed as the space required to store all base models and the meta learner, while inference time complexity for a stacking ensemble is computed as the time taken to

generate predictions from all base models and the time taken by the meta learner to combine these predictions. Space and inference time complexity can be measured as follows (*Zhou, 2012*; *Tsoumakas, Partalas & Vlahavas, 2008*):

$$\text{Space Complexity} = O(N \cdot S_{base} + m \cdot N + S_{meta}) \tag{1}$$

where $N$ is the number of base models in the stacking ensemble, $S_{base}$ is the space required to store each base model, $m$ is the number of samples, and $S_{meta}$ is the space required to store the meta learner.

$$\text{Time Complexity} = O(N \cdot T_{predict} + T_{meta\_predict}) \tag{2}$$

where $N$ is the number of base models in the stacking ensemble, $T_{predict}$ is the time complexity for making a prediction with each base model and $T_{meta\_predict}$ is the time complexity for the meta learner to make the final prediction.

Stacking ensemble complexity will be measured based on the number of the base models needed to construct the ensemble (*Chatzimparmpas et al., 2020*). Specifically, we will evaluate how effectively GS can quickly identify a subset of high performing base models, and how BE can fine-tune this subset by eliminating base models that do not contribute significantly to the ensemble performance. Both strategies aim to balance computational efficiency with effectiveness, ultimately reducing the overall ensemble complexity in terms of both the number of base models and the computational resources required for training and inference. In our empirical study, we will empirically evaluate to which extent DSEs can achieve efficient and effective code smell detection across programming languages, offering a practical solution for maintaining software quality in multi-language development environments.

## RESEARCH METHODOLOGY

The main objective of this empirical study is to investigate the complexity and detection performance stability of our proposed dynamic stacking ensembles in code smells detection across Java and Python programming languages code smells. Our study goal is formulated as follows: **evaluate** dynamic stacking ensembles built using the greedy search and backward elimination strategies for the **purpose** code smell detection with **respect** to their detection performance measured in accuracy, F1-score, and AUC scores from the **perspective** of both researchers and software engineers within the **context** of four Java and two Python code smells datasets. We have formulated the following research questions to achieve our research objective:

• **RQ 1.** Will greedy search and backward elimination dynamic ensembles strategies yield to two different base models lists?

**Rationale**. We will investigate if both dynamic ensembles creation strategies will result in different base models. We will contrast between the two strategies in terms of size and the type of selected base models used to detect each investigated Java and Python code smell.

• **RQ 2.** Will dynamic ensembles have stable code smell detection performance across different languages in comparison to all candidate base models?

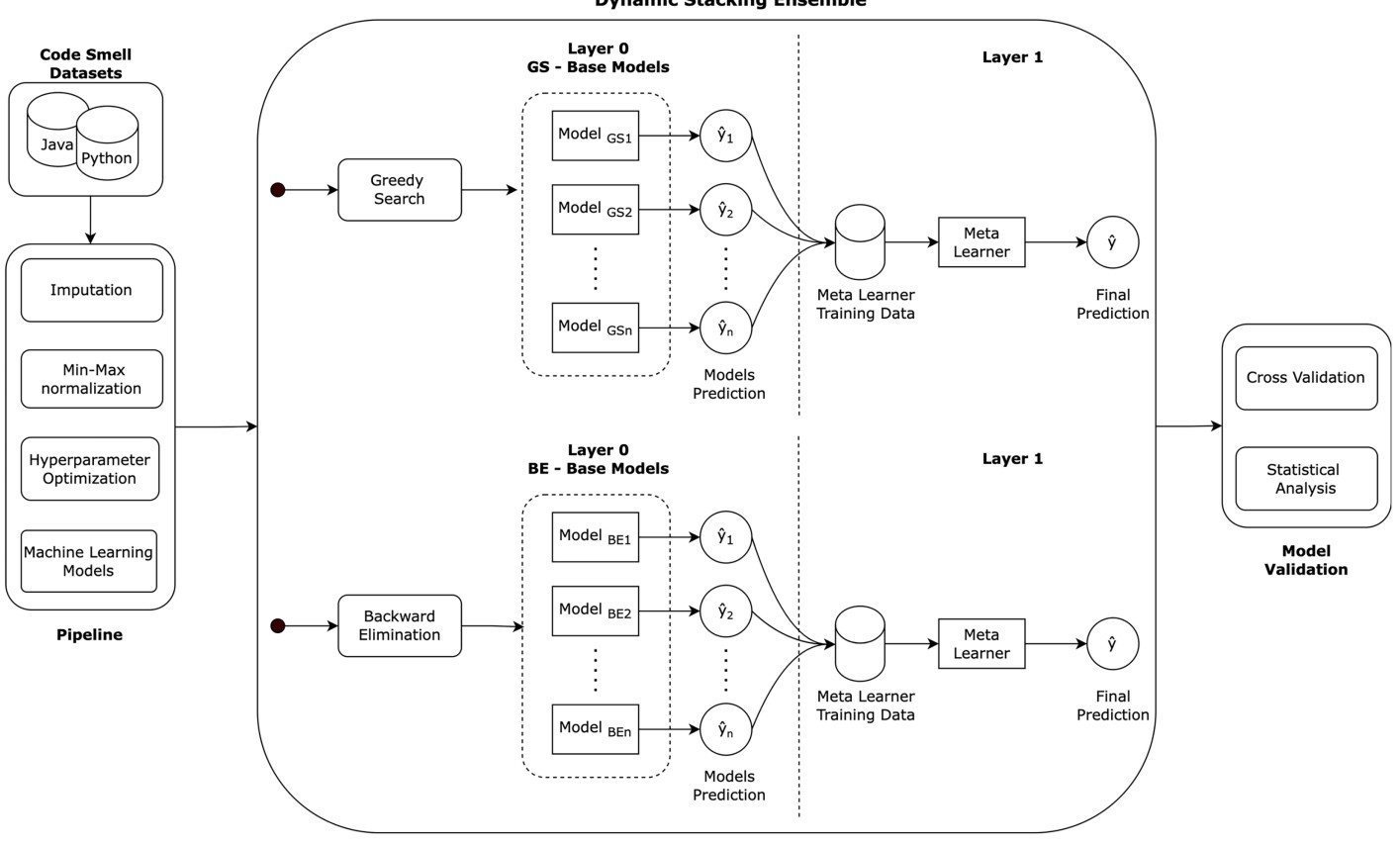

**Figure 2  Research methodology.**

**Rationale**. Greedy search and backward elimination will result in different base models to build the dynamic stacking ensembles. In this research question, we aim to examine the detection performance stability of dynamic ensembles against all candidate base models (*i.e.*, both selected and eliminated base models) in detecting Java and Python code smells

• **RQ 3.** Will dynamic stacking ensembles result in less complex ensembles with comparable detection performance to full stacking ensembles?

**Rationale**. One of study goals is to empirically prove that we can build less complex dynamic ensembles without sacrificing significant detection performance. In this research question, we will contrast the dynamic ensembles complexity against full stacking ensembles, and to which degree there will be a detection performance loss.

Figure 2 illustrates the followed research methodology of the dynamic stacking ensemble approach in cross-language code smell detection. The methodology consists of several key stages. First, each Java and Python code smell dataset undergoes a data pre-processing pipeline: imputation, normalization, and models hyperparameters optimization. Then, dynamic stacking ensembles are created using the greedy search and backward eliminations algorithms. In greedy search, the stacking ensemble base models

are selected incrementally based on the model that is locally optimal at each step. On the other hand, in backward elimination, the stacking ensemble base models are the result of full base models list, and then incrementally removing the least contributing model. Finally, the approach is validated using the cross validation method, and statistical analysis is employed to analyze the models detection performance ensuring the reliability of our performance assessment.

## EMPIRICAL STUDY DESIGN

In this section, we will discuss the details of our empirical study design details and any choices that were made before the execution of the study. The empirical study was fully implemented using Python programming language utilizing the Scikit-Learn library to build all the machine learning models. Our code is made open source with a replication package uploaded in Zenodo (https://doi.org/10.5281/zenodo.11488752).

### Code smell datasets

The primary objective of this study is to investigate the performance and stability of dynamic stacking ensembles in detecting code smells across different programming languages. Therefore, the models were trained and evaluated on datasets from two programming languages: Java and Python. Table 5 shows a summary of the used datasets. Each code smell dataset was pre-processed in a pipeline consisting of two steps: missing data imputation and feature scaling. In data imputation, we imputed the missing data in the feature with the mean values in that feature. Next, we scaled the features using Min-Max normalization into values between [0,1]. Both steps will assist us in producing robust and faster models (*Acuna & Rodriguez, 2004*; *Singh & Singh, 2020*). All the pre-processing steps were performed within a pipeline and using the training data only, to prevent data leakage.

#### Java dataset

*Arcelli Fontana et al. (2016)* introduced Java datasets (https://essere.disco.unimib.it/machine-learning-for-code-smell-detection) containing 2,520 samples of two class-level code smells (God Class and Data Class) and four method-level code smells (Long Method, Feature Envy, Long Parameter List, and Switch Statements). These datasets contain 420 samples for each code smell type, where the independent variables are Object Oriented (OO) metrics and the dependent variable is a class indicating whether the sample is smelly or non-smelly.

#### Python dataset

*Sandouka & Aljamaan (2023)* developed a recent open access Python datasets (https://doi.org/10.5281/zenodo.7512516) for code smell detection. This dataset contains 2,000 samples of a class-level (Large Class) and a method-level (Long Method) code smells. The independent variables are classified into two groups: raw metrics that do not need complex calculations (*e.g.*, number of code lines) and Halstead complexity metrics to extract a quantitative complexity measures (*e.g.*, program vocabulary). The dependent variable is a binary value: smelly or non-smelly.

**Table 5 Datasets distribution.**

| Dataset | Level | Code smell | Independent variables | Number of samples | | |
| --- | --- | --- | --- | --- | --- | --- |
| | | | | Smelly | Non-smelly | Total |
| *Arcelli Fontana et al. (2016)* | Class | God class | 61 OO metrics | 140 | 280 | 420 |
| | | Data class | | | | |
| | Method | Feature envy | 82 OO metrics | | | |
| | | Long method | | | | |
| | | Long parameter list | 55 OO metrics | | | |
| | | Switch statements | | | | |
| *Sandouka & Aljamaan (2023)* | Class | Large class | 18 code metrics | 200 | 800 | 1,000 |
| | Method | Long method | | | | |
| **Total** | | | | 1,240 | 3,280 | 4,520 |

### Code smell definition

Code smells were introduced by *Fowler (1999)* back in 1999, with refactoring strategies to counter them. Code smells were defined and categorized according to their granularity (*e.g.*, class-level, method-level). In this article, we are targeting class-level and method-level code smells from two different programming languages: Java and Python. These smells are defined as follows (*Fowler, 1999*):

- **God Class (GC):** is a class-level code smell indicating a class with many responsibilities and encapsulating most the software functionalities.
- **Data Class (DC):** is a class-level code smell representing a class that stores data by having a large number of attributes with accessors.
- **Large Class (LC):** is a class-level smell indicating a class that is bloated with many fields, methods, and lines of code.
- **Long Method (LM):** is a method-level code smell representing a method with many lines of code with multiple functionalities making it difficult to read and maintain.
- **Long Parameter List (LPL):** is a method-level code smell that occurs when a method have too many parameters relatively to other methods in the same class.
- **Feature Envy (FE):** is a method-level code smell representing a method that extensively used other classes attributes and methods, more than its own class.
- **Switch Statements (SS):** is a method-level code smell that occurs when your code has complex switch statements or a series of if-conditions.

The rationale behind selecting the above code smells is driven by three main factors that highlight their impact on software quality (*Al-Shaaby, Aljamaan & Alshayeb, 2020*; *Kaur, 2020*; *Alazba, Aljamaan & Alshayeb, 2023*; *Yang et al., 2022*), as follows: (i) they have significant impact on software maintainability and quality, thus, their detection is important for improved code quality. (ii) They represent smells from different levels of granularity, *i.e.* class-level and method-level smells, providing diverse set of smells. (iii)

They impose different challenges in their detection by machine learning models due to their unique characteristics. (iv) These smells are the most investigated smells in the literature and can provide a benchmark performance for our models.

## Stacking ensembles

In this section, we will overview the six models that were selected as candidate base models to build our dynamic stacking ensembles. These models were selected from different classification families to diverse the candidate base models list. Next, we will explain the hyperparameters tuning process followed to tune the base models.

### Base models

In this section, we briefly present a description of each base model used in this research.

**Decision tree (DT).** It is one of the powerful ML algorithms for both classification and regression tasks. One of the most prevalent and widely used training algorithms for constructing decision trees is the Classification and Regression Trees (CART) algorithm, which is implemented by Scikit-Learn library. In binary classification, CART finds the optimal data splits at each internal node to divide each set using Gini Index impurity. This process is accomplished by using the Gini Index impurity metric, that is calculated as *Quinlan (2014)*:

$$GI = 1 - \sum_{i=0}^{2} p_i^2 \qquad (3)$$

where $p$ is the probability of class $i$. The splitting procedures is recursively repeated until the algorithm reaches a maximum depth or if impurity cannot be further reduced.

**K-nearest neighbors (KNN).** It is characterized by its extreme simplicity and its unique attribute of not requiring training the data. The core of KNN is the underlying assumption that similar datapoints are spatially positioned next to each other within the space. It uses the whole training samples to represent the model and predicts the label of a new sample by finding the nearest k neighbors (samples) using the Euclidean distance, as *Peterson (2009)*:

$$d(p, q) = \sqrt{\sum_{i=1}^{n} (p_i - q_i)^2} \qquad (4)$$

where $d(p, q)$ is the distance between datapoints $p$ and $q$, $p_i$ and $q_i$ are the vectors values of the two datapoints, and $n$ is the space dimensions.

**Logistic regression (LR).** It is a statistical model primarily designed for binary classification tasks, in which a given data point is assigned to one of two mutually exclusive classes. Its core idea is calculating the probability of each datapoint to be classified to one class. The algorithm sets a threshold value, typically 0.5, upon which it bases its final classification decision. Consequently, datapoints with probabilities exceeding the threshold

are assigned to one class, while those falling below it are assigned to the other class. The probability value is calculated as *Hilbe (2009)*:

$$\hat{p} = \sigma(WX + B) \tag{5}$$

where $X$ is the input features, $B$ is a bias value, $W$ is the weights of the model, and $\sigma$ is the logistic sigmoid function.

**Multi-layer perceptron (MLP).** It is Feedforward Neural Network (FFN) that is fully connected. It is composed of an input layer, one (or more) hidden layer(s), and an output layer. Each layer feeds its output to the next layer until the last layer is reached. Upon reaching the final output layer, the backpropagation algorithm is initiated, a crucial phase in the neural network training process. During this phase, the cost function is computed to guide the iterative adjustment of the network's weights. This iterative optimization process aims to minimize the cost function, ultimately leading to maximizing the model's performance (*Haykin, 1998*).

**Naive Bayes (NB).** It is a probabilistic models that is grounded in the principles of Bayes' theorem, expressed as Eq. (6). The term "naive" signifies the model's assumption that the features are completely independent. We used the Bernoulli variant of NB as it is more suitable for data characterized by binary features (*Rish, 2001*).

$$P(A|B) = \frac{P(B|A) * P(A)}{P(B)} \tag{6}$$

**Support vector machine (SVM).** It is considered as one of the most popular and versatile ML algorithms. It can be used to address different machine learning tasks, including both linear and nonlinear classification, regression analysis, as well as outliers' detection. SVM uses the training data with objective of identifying the optimal hyperplane that can effectively separates the datapoints into their respective classes. To learn the optimal line, SVM finds the closest points (*i.e.*, support vectors) of each class and calculates the distance (*i.e.*, margin) between the line and the support vectors, such that the margin is maximized (*Vapnik, 2013*).

### Hyperparameters optimization

In training machine learning models, hyperparameters optimization holds profound rule in a model's behavior and performance, affecting factors such as accuracy, generalization, and training efficacy (*Hoque & Aljamaan, 2021*). It ensures that machine learning models are optimized for a given task, maximizing their potential and utility.

In this research, we utilized Optuna (*Akiba et al., 2019*), a powerful and versatile Python library used for automating the hyperparameters tuning process of machine learning models. It efficiently explores the hyperparameters space and adjusts the hyperparameters values to maximize the model's performance metrics, typically using cross-validation.

In our comprehensive model training and optimization process, we conducted hyperparameters tuning for each individual base model. The hyperparameters space is shown in Table 6, where round brackets "()" indicate a value selected from a range with a

**Table 6 Hyperparameters space of the optimization process.**

| Model | Hyperparameter | Values |
| --- | --- | --- |
| DT | Max depth | (2–12) |
| | Splitter | {best, random} |
| | Max features | {None, sqrt, log2} |
| KNN | Weights | {uniform, distance} |
| | Metric | {euclidean, manhattan, minkowski} |
| | Neighbors | (1–20) |
| LR | Penalty | {11, 12} |
| | C | (0–100) |
| MLP | Activation | {relu, identity, logistic,tanh} |
| | Solver | {adam, lbfgs, sgd} |
| | Alpha | (0.0001–100) |
| | Learning rate | {constant, invscaling, adaptive} |
| NB | Alpha | (0.0001–100) |
| | binarize | (0.0–10.0) |
| SVM | Kernel | {rbf, linear, poly, sigmoid} |
| | Gamma | {scale, auto} |
| | C | (0.001–100) |

minimum and maximum value, and curly brackets "{}" indicate a choice selected from the available options.

## Model validation

Stacking ensembles and base models were validated using 10 stratified folds cross validation repeated 10 times. In this validation approach, the dataset is divided into 10 stratified folds; nine folds for training and the remaining fold for testing. The approach will be repeated 10 times, where each fold will be used for testing exactly once. This whole approach will be repeated again 10 times, where the model performance estimate will be the average of 100 testing sets. With this 10 repetition of the 10 fold cross validation, we ensure that we produce unbiased model estimates with low variance (*Tantithamthavorn et al., 2016*).

## Evaluation measures

The purpose of this study is to investigate cross-language code smells detection. This problem is considered as a binary classification problem, where the ML models take a code instance and output one of two classes: smelly or non-smelly code. Two threshold-dependent evaluation metrics (Accuracy and F1-score) were employed, along with one threshold-independent metric (AUC).

### Accuracy

Accuracy (*Witten et al., 2005*) stands out as one of the most commonly used metrics for binary classification problems. Therefore, it is an essential metric in evaluating the

performance of code smells detection models. Accuracy is defined as the percentage of correctly classified instances (*i.e.* true positive (TP) and true negative (TN) predictions) over all samples (*i.e.* including false positive (FP) and false negative (FN) predictions), as shown in Eq. (7).

$$Accuracy = \frac{TP + TN}{TP + TN + FP + FN} \times 100 \qquad (7)$$

### F1-score

F1-score is calculated as the harmonic average of precision and recall, assigning equal importance to both. Precision uses TPs and FPs to find how many of the detected code smells are correct, whereas recall uses TPs and FNs to find how many code smells are detected correctly. Eqs. (8)–(10) show how precision, recall, and F1-score are calculated (*Witten et al., 2005*), respectively.

$$Precision = \frac{TP}{TP + FP} \qquad (8)$$

$$Recall = \frac{TP}{TP + FN} \qquad (9)$$

$$F1{-}score = 2 \times \frac{Precision \times Recall}{Precision + Recall}. \qquad (10)$$

### AUC

Area Under the Curve (AUC) (*Witten et al., 2005*) is a threshold-independent metric that provides a better understanding of the models performance than threshold-dependent metrics. It represents the percentage of the area underneath the Receiver Operator Characteristic (ROC) curve, that plots the relationship between the TP and FP rates. AUC values ranges from 0 to 1, with a higher value indicating a better performance.

## Statistical analysis

A statistical test was conducted to examine whether the performance of the dynamic stacking ensemble approaches significantly differ from the performance of full stacking ensemble and the six base models (*Demšar, 2006*). This empirical study has a total of 576 comparisons (72 pairwise comparisons per dataset × eight datasets). We used the F1-score metric resulting from 100 runs (10 folds CV repeated 10 times) as it is the most used metric in binary classification problems and it combines both precision and recall. The non-parametric Wilcoxon signed-rank test is used to perform the pairwise comparisons due to the independence of the samples. Moreover, Wilcoxon signed-rank test is more fixable since it does not require the data to be normally distributed. The tests are performed with 95% confidence interval (*i.e.*, at a significance level $\alpha = 0.05$). For each pairwise comparison, the *p*-value will be used to accept/reject the following hypotheses:

- $H_0 : F1{-}Score_x = F1{-}Score_y$ (there is no difference in the detection performance between the two models)

**Table 7 DSE base models for each code smell.**

| Dataset | Ensemble | Base model | Size |
|---------|----------|-----------|------|
| (J) GC | DSE-GS | MLP, DT, SVM, KNN, LR, NB | 6 |
|        | DSE-BE | KNN, LR | 2 |
| (J) DC | DSE-GS | KNN, LR | 2 |
|        | DSE-BE | KNN, DT, SVM | 3 |
| (J) LM | DSE-GS | DT, SVM, KNN | 3 |
|        | DSE-BE | KNN, LR | 2 |
| (J) LPL | DSE-GS | DT, LR, MLP, KNN, NB, SVM | 6 |
|         | DSE-BE | DT, SVM | 2 |
| (J) FE | DSE-GS | KNN, SVM, DT, NB | 4 |
|        | DSE-BE | DT, SVM | 2 |
| (J) SS | DSE-GS | KNN, SVM, LR, DT, MLP, NB | 6 |
|        | DSE-BE | KNN, LR, NB | 3 |
| (P) LC | DSE-GS | LR, DT, MLP | 3 |
|        | DSE-BE | DT, LR | 2 |
| (P) LM | DSE-GS | KNN, DT, MLP, SVM, LR, NB | 6 |
|        | DSE-BE | KNN, DT, LR, SVM, MLP, NB | 6 |

- $H_1 : F1-Score_x \neq F1-Score_y$ (there is a significant difference in detection performance between the two models)

Nevertheless, multiple statistical tests on the same dataset leads to an increase of Type I error. To reduce this error, we followed Bonferroni correction that adjusts the *p*-value as follows:

$$adj.p-value = \frac{1 - (1 - \alpha)}{T} \qquad (11)$$

where $\alpha$ is the old *p*-value and $T$ is the number of the performed comparisons.

## RESULTS AND DISCUSSIONS

This section discusses our empirical study's results and addresses the previously formulated research questions.

### Dynamic stacking ensembles algorithms

This section answers the first formulated research question: "*Will greedy search and backward elimination dynamic ensembles strategies yield to two different base models lists?*". We constructed our DSE using greedy search and backward elimination strategies to detect each investigated code smell, and the resulting base models for each DSE are listed in Table 7. It can be observed that both strategies to build the DSE resulted in different base model lists. Each DSE had varying base model list sizes with different combinations.

Figure 3 presents a summary of base models lists created *via* greedy search and backward elimination strategies based on code smell and base model type. We can observe

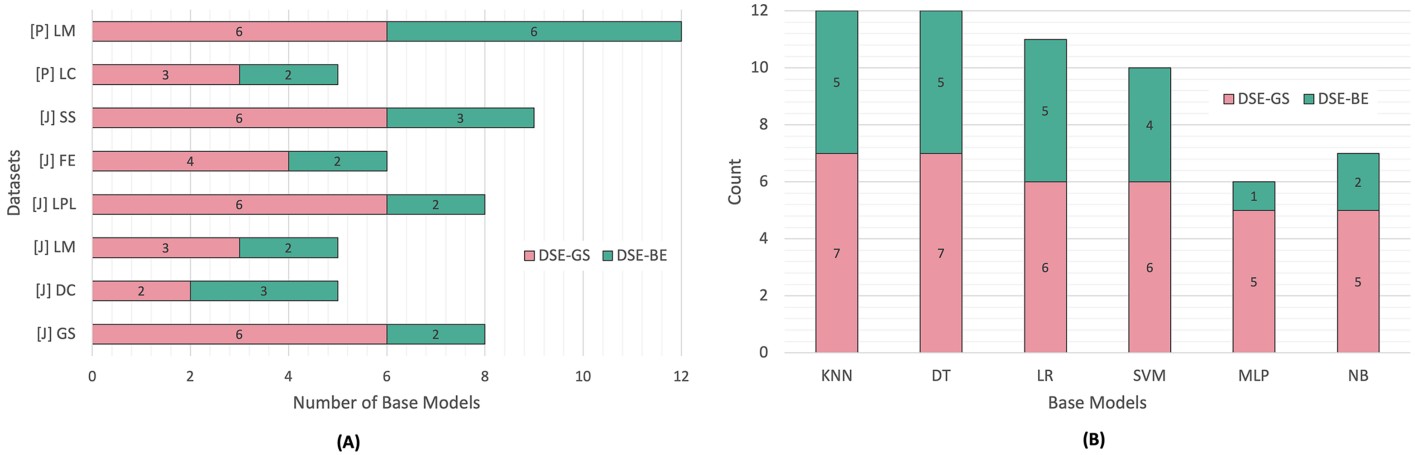

**Figure 3** **DSE base models summary.** (A) Per code smell. (B) Per base model.

that the base models list sizes varied based on the code smell detected. In Python code smells detection, the LM code smell was the most challenging smell to detect, since the resulting base models for both DSEs were equivalent to an FSE (*i.e.*, six base models), while the LC code smell required a smaller base model list in both dynamic ensembles.

Similarly, In Java code smells detection, different code smells resulted in different base models lists for both DSE strategies. The SS code smell was the most challenging smell to detect, while the DC code smell, required fewer base models. After comparing the DSE strategies, we can conclude that backward elimination strategy resulted in smaller base models list as compared to the greedy search strategy. In fact, in 50% of the code smells, base models created *via* the greedy search strategy were equivalent to an FSE. Figure 3 presents another perspective regarding base models selection count per DSE. We can observe that NB and MLP models were the least favored models in relation to both DSE strategies, while the KNN and DT models were the most commonly selected base models by both DSE strategies.

> **RQ1 Answer.** Greedy search and backward elimination strategies yielded different base models lists to build dynamic ensembles for code smell detection. It was notable that backward elimination strategy to build the dynamic ensembles resulted in smaller base model lists compared to the greedy search strategy. Among the six base models, KNN and DT models were favored to be selected as base models in both greedy search and backward elimination strategies, while MLP and NB models where the least selected base models.

## Dynamic stacking ensemble *vs*. base models

This section addresses the second formulated research question: "*Will dynamic ensembles have stable code smell detection performance across different languages in comparison to all*

**Table 8 Code smell detection performance for stacking ensembles and base models.**

| Classifiers | (J) Data class | | | (J) Feature envy | | | (J) God class | | | (J) Long method | | |
|---|---|---|---|---|---|---|---|---|---|---|---|---|
| | Acc. | F1-score | AUC | Acc. | F1-score | AUC | Acc. | F1-score | AUC | Acc. | F1-score | AUC |
| KNN | 94.5 | 92.56 | 0.99 | 89.9 | 83.2 | 0.96 | 93.81 | 89.85 | 0.98 | 93.21 | 88.88 | 0.99 |
| DT | 98.81 | 98.24 | 0.99 | 96.38 | 94.73 | 0.97 | 95.67 | 93.42 | 0.96 | 98.83 | 98.26 | 0.99 |
| LR | 97.55 | 96.43 | 0.99 | 93.86 | 90.6 | 0.98 | 96.33 | 94.37 | 0.99 | 98.93 | 98.41 | 1 |
| SVM | 96.33 | 94.7 | 0.99 | 94.33 | 91.38 | 0.97 | 96.36 | 94.28 | 0.99 | 96.81 | 95.26 | 1 |
| MLP | 93.21 | 90.05 | 0.98 | 93.02 | 89.09 | 0.98 | 93.1 | 88.43 | 0.99 | 93.81 | 89.75 | 0.99 |
| NB | 91.31 | 87.02 | 0.97 | 85 | 81.13 | 0.91 | 92.86 | 90.1 | 0.98 | 85.69 | 81.34 | 0.92 |
| FSE | 98.71 | 98.06 | 1 | 96.21 | 94.42 | 0.99 | 96.74 | 95.1 | 1 | 99.14 | 98.69 | 1 |
| DSE-GS | 98.71 | 98.04 | 1 | 96.26 | 94.48 | 0.99 | 96.81 | 95.21 | 0.99 | 99.29 | 98.91 | 1 |
| DSE-BE | 98.79 | 98.16 | 1 | 96.17 | 94.29 | 0.99 | 96.71 | 95.09 | 0.99 | 99.19 | 98.78 | 1 |
| Classifiers | (J) Long parameter list | | | (J) Switch statements | | | (P) Large class | | | (P) Long method | | |
| | Acc. | F1-score | AUC | Acc. | F1-score | AUC | Acc. | F1-score | AUC | Acc. | F1-score | AUC |
| KNN | 74.52 | 43.63 | 0.8 | 80.76 | 63.44 | 0.88 | 90.63 | 71.59 | 0.92 | 91.48 | 79.25 | 0.97 |
| DT | 91.67 | 87.01 | 0.95 | 84.83 | 75.04 | 0.91 | 89.96 | 71.74 | 0.87 | 94.77 | 89.05 | 0.97 |
| LR | 91.05 | 85.86 | 0.97 | 88.07 | 79.58 | 0.95 | 91.89 | 76.6 | 0.94 | 91.45 | 80.03 | 0.96 |
| SVM | 88.64 | 79.42 | 0.97 | 85.21 | 73.84 | 0.93 | 92.81 | 79.55 | 0.94 | 93.51 | 85.63 | 0.98 |
| MLP | 76.02 | 42.11 | 0.91 | 84.86 | 72.7 | 0.93 | 92.59 | 79.55 | 0.94 | 78.96 | 18.85 | 0.89 |
| NB | 61.48 | 47.52 | 0.67 | 70.24 | 63.23 | 0.79 | 76.8 | 51.88 | 0.79 | 72.63 | 47.24 | 0.75 |
| FSE | 91.69 | 86.67 | 0.92 | 87.79 | 79.08 | 0.95 | 92.73 | 79.72 | 0.94 | 95.08 | 89.39 | 0.98 |
| DSE-GS | 91.86 | 87.12 | 0.93 | 87.71 | 79.14 | 0.95 | 92.74 | 79.74 | 0.94 | 95.07 | 89.4 | 0.98 |
| DSE-BE | 91.95 | 87.21 | 0.93 | 87.45 | 78.57 | 0.95 | 92.86 | 80.09 | 0.94 | 95.04 | 89.34 | 0.99 |

*candidate base models?"*. After constructing our DSE, we will examine their code smell detection stability against all candidate base models across all code smells. Table 8 presents the models detection performance in terms of accuracy, F1-score, and AUC scores in detecting Java and Python code smells.

For Java code smells detection, base models struggled to achieve a stable and consistently high detection performance across all Java code smells. Base models were able to achieve high detection performance in detecting DC and GC code smells, but had far lower detection performance in more challenging smells, such as: LPL and SS code smells. For instance, KNN achieved an F1-score of 92% in detecting DC, but the performance was degraded significantly to 43% when the model was used to detect LPL smells. Similarly, in Python code smells detection, base models struggled to achieve stable and high detection performance in detecting both Python code smells. Rather, stacking ensembles both (DSE-GS and DSE-BE) exhibited a stable and consistently high detection performance for Java and Python smells in comparison to all base models.

To further examine the detection performance distribution, we plotted the F1-scores boxplots for dynamic ensembles and base models for Java and Python code smells, as illustrated in Fig. 4. The base models demonstrated varying levels of performance in

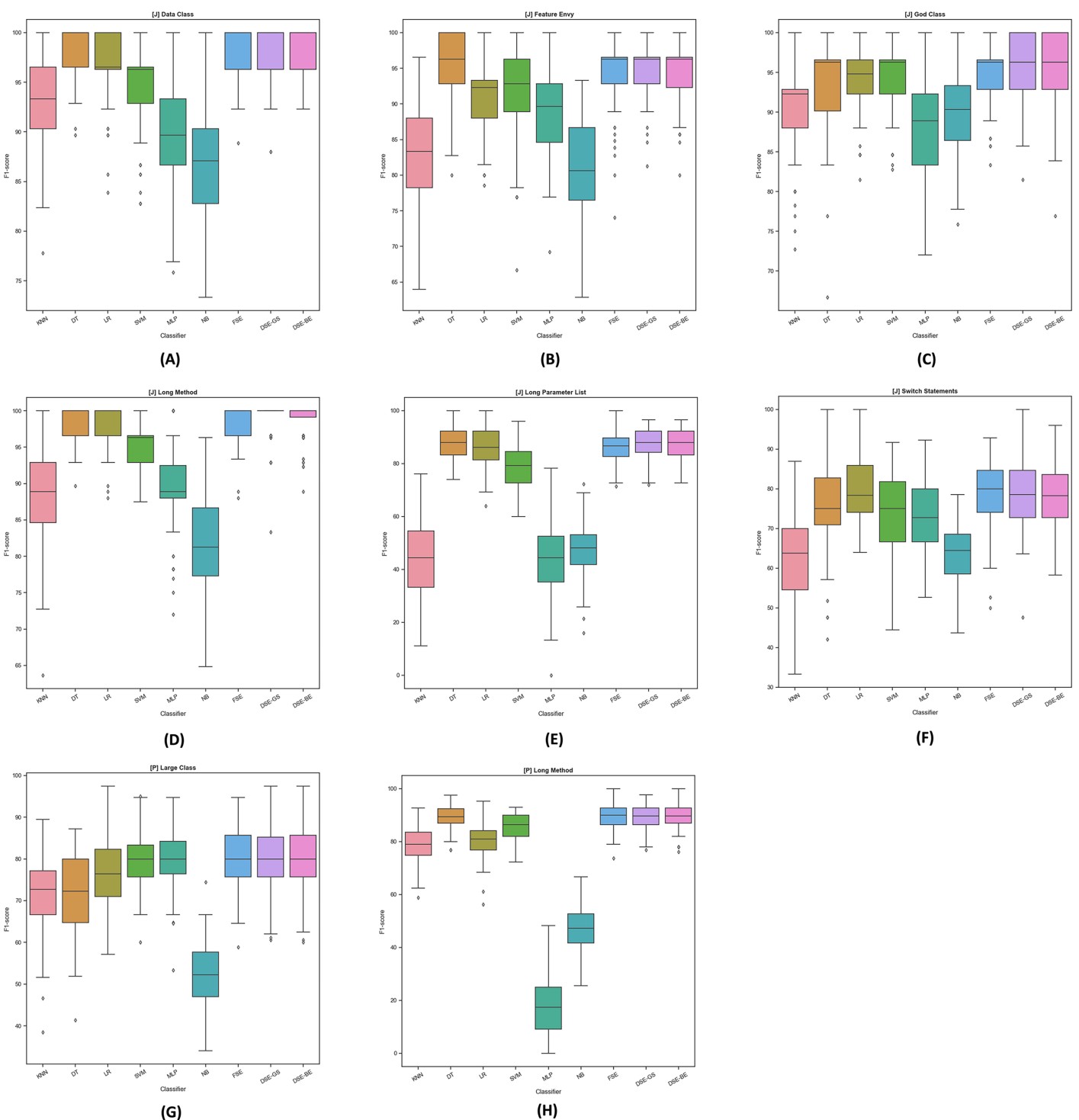

**Figure 4 F1-score boxplots for stacking ensembles and base models performance per code smell.** (A) (Java) Data Class. (B) (Java) Feature Envy. (C) (Java) God Class. (D) (Java) Long Method. (E) (Java) Long Parameter List. (F) (Java) Switch Statement. (G) (Python) Large Class. (H) (Python) Long Method.

**Table 9 Models pairwise comparison results.**

| Classifier | SVM | NB | MLP | LR | KNN | DT | FSE | DSE-GS | DSE-BE | # Wins | % Wins |
|---|---|---|---|---|---|---|---|---|---|---|---|
| SVM | – | 8 | 5 | 1 | 7 | 1 | 0 | 0 | 0 | 22 | 34% |
| NB | 0 | – | 1 | 0 | 0 | 0 | 0 | 0 | 0 | 1 | 2% |
| MLP | 0 | 5 | – | 0 | 3 | 1 | 0 | 0 | 0 | 9 | 14% |
| LR | 3 | 8 | 6 | – | 7 | 2 | 0 | 0 | 0 | 26 | 41% |
| KNN | 0 | 4 | 2 | 0 | – | 0 | 0 | 0 | 0 | 6 | 9% |
| DT | 5 | 8 | 6 | 3 | 7 | – | 0 | 0 | 0 | 29 | 45% |
| FSE | 6 | 8 | 7 | 4 | 8 | 1 | – | 0 | 0 | 34 | 53% |
| DSE-GS | 6 | 8 | 7 | 3 | 8 | 1 | 0 | – | 0 | 33 | 52% |
| DSE-BE | 6 | 8 | 7 | 2 | 8 | 1 | 0 | 0 | – | 32 | 50% |
| # Losses | 26 | 57 | 41 | 13 | 48 | 7 | 0 | 0 | 0 | | |
| % Losses | 41% | 89% | 64% | 20% | 75% | 11% | 0% | 0% | 0% | | |

detecting Java code smells. For instance, in LPL code smell detection, base models exhibited perceptible variations in detecting this code smell; conversely, dynamic ensembles exhibited higher detection performance. In addition, for Python code smells detection, base models failed to maintain high detection rates across both smells; for example, the MLP model detection varied significantly between LC and LM Python code smells detection. Overall, we can observe the stability of dynamic ensembles in detecting Java and Python code smells by having their boxes in the top right corner above most of the base models boxes. In addition, dynamic ensembles had smaller boxes and shorter whiskers, thus confirming their detection performance stability.

We performed a statistical pairwise comparison to examine whether the observed detection performance was statistically significant or not. The results from the models pairwise comparison are presented in Table 9. Each model has a total of 64 pairwise comparisons against eight other models to detect eight code smells. This table can be read row and column wise. When reading it row wise, the number in the table indicates the number of row model wins against the column model, and *vice versa*, when reading it column wise, the number indicates the column model losses against the row model. Each pairwise comparison will have three possible outcomes: (1) win, (2) loss, or (3) tie, meaning that the difference is insignificant.

For example, the DSE-GS model as an example. It shows that this dynamic ensemble won against the SVM model in six comparisons (*i.e.*, code smells); this can also be interpreted as indicating the loss of the SVM model lost against DSE-GS in six comparisons. The table outcomes confirms our previous findings that base models had varying detection performances based on winning and losing in relation to code smells detection. Conversely, dynamic ensembles had the highest percentage of wins and never had a single loss against any base model. Previously reported pairwise comparisons can also be viewed as well per code smell, as presentred in Table 10. Different numbers of wins and losses have been reported per code smell, indicating the models' different challenges in detecting them. Python LM code smell was the most challenging smell to detect.

**Table 10 Models results per code smell.**

| Classifier | (J) DC | | (J) FE | | (J) GC | | (J) LM | | (J) LPL | | (J) SS | | (P) LC | | (P) LM | |
|---|---|---|---|---|---|---|---|---|---|---|---|---|---|---|---|---|
| | W | L | W | L | W | L | W | L | W | L | W | L | W | L | W | L |
| SVM | 2 | 4 | 2 | 4 | 3 | 0 | 3 | 5 | 3 | 5 | 2 | 4 | 3 | 0 | 4 | 4 |
| NB | 0 | 8 | 0 | 7 | 0 | 6 | 0 | 8 | 0 | 6 | 0 | 7 | 0 | 8 | 1 | 7 |
| MLP | 1 | 7 | 2 | 4 | 0 | 6 | 1 | 6 | 0 | 6 | 2 | 4 | 3 | 0 | 0 | 8 |
| LR | 3 | 3 | 2 | 4 | 3 | 0 | 4 | 0 | 4 | 0 | 5 | 0 | 3 | 1 | 2 | 5 |
| KNN | 2 | 5 | 0 | 7 | 0 | 6 | 1 | 6 | 0 | 6 | 0 | 7 | 1 | 6 | 2 | 5 |
| DT | 5 | 0 | 5 | 0 | 3 | 0 | 4 | 0 | 4 | 0 | 2 | 1 | 1 | 6 | 5 | 0 |
| DSE-GS | 5 | 0 | 5 | 0 | 3 | 0 | 4 | 0 | 4 | 0 | 4 | 0 | 3 | 0 | 5 | 0 |
| DSE-F | 5 | 0 | 5 | 0 | 3 | 0 | 4 | 0 | 4 | 0 | 4 | 0 | 4 | 0 | 5 | 0 |
| DSE-BE | 4 | 0 | 5 | 0 | 3 | 0 | 4 | 0 | 4 | 0 | 4 | 0 | 3 | 0 | 5 | 0 |
| Total | 27 | | 26 | | 18 | | 25 | | 23 | | 23 | | 21 | | 29 | |

**RQ2 Answer.** Base models showed varying detection performance in detecting Java and Python code smells. DT and LR models were the highest performing base models, while KNN and NB models were the lowest performing models. Both dynamic ensembles (DSE-GS and DSE-BE) have proven to be the most stable models in detecting cross language smells by achieving the highest detection performance and not losing to any base model.

## Dynamic stacking *vs*. full stacking

This section answers the third formulated research question: "*Will dynamic stacking ensembles result in less complex ensembles with comparable detection performance to full stacking ensembles?*" DSEs were proven to be stable models in term of detecting code smells over all base models. Below, we will contrast the the DSE against the FSE in terms of complexity and detection performance.

Figure 5 contrasts the FSE and DSE ensembles base models size across Java and Python code smells. The FSEs were built using all six base models, while, the DSEs had varying base models sizes for each code smell. DSE-GS had an equal base models size relative to the FSE for four smells, while exhibiting a lower size in the other four smells. Conversely, DSE-BE had lower base models sizes than FSE for seven code smells, and only one smell required DSE-BE for the use of the full base models. Overall, we can conclude that for most code smells, DSE resulted in less complex models in comparison to FSE.

Detection performances differences between FSE and DSE were marginal, as outlined in Table 8. FSEs were not able to perform significantly higher than DSEs in detecting any of the Java and Python code smells, as confirmed in Table 9. Moreover, as indicated in Fig. 4 boxplots of both FSE and DSE exhibit similar sizes and whiskers lengths, with no noticeable differences.

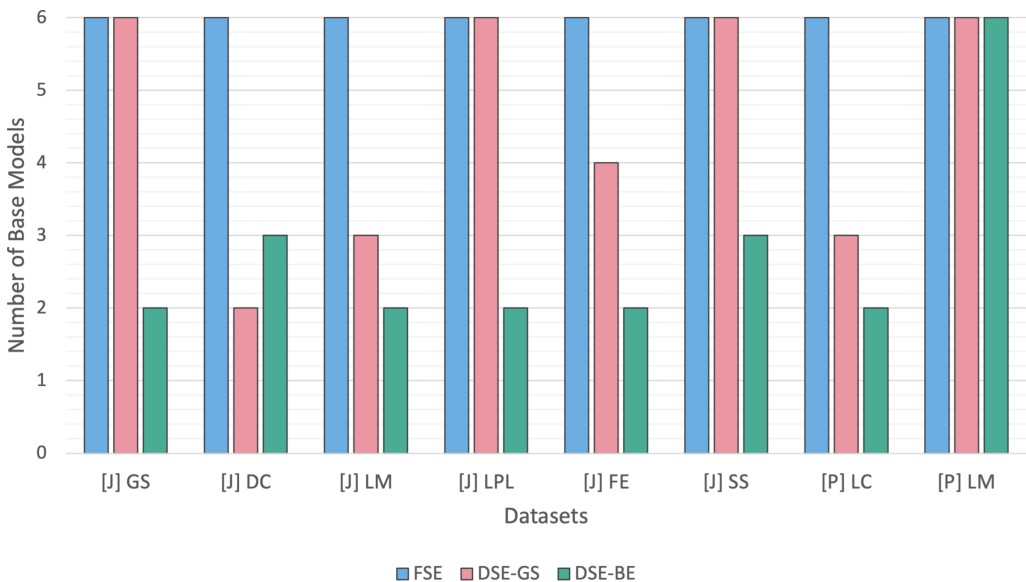

**Figure 5  Dynamic stacking ensembles size against full stack.**

**RQ3 Answer.** Dynamic ensembles in comparison to full stacking ensembles yielded less complex models in terms of detecting most of the investigated code smells, with the backward elimination strategy resulting in less complex models. Dynamic ensembles were able to perform comparably against full stacking ensembles and there was no significant detection difference between them in detecting any code smell.

# THREATS TO VALIDITY

In this section, we will discuss potential threats to validity of our study and the measures taken to mitigate them. By discussing these threats, we aim to provide a transparent and reliable assessment of our study findings. An internal threat to validity might rise from our implementation choices in building the machine learning models. To mitigate this risk, we adhered to standard programming practices in building our machine learning models, and used well known machine learning libraries to perform our study. Scikit-Learn library was utilized to build all machine learning models, while Optuna was used to automate the models hyperparameters tuning process. Moreover, our code is open sourced and publicly available making our results replicable and verifiable by other researchers. Another internal threat to validity is related to our models evaluation metrics. The choice of evaluation metrics could introduce a bias in our results. To mitigate this threat, we used a comprehensive set of metrics, including accuracy, auc, and F1-score to provide a balanced assessment of our models detection performance.

An external threat to validity involves the generalizability of our study findings. This can be effected by the targeted programming languages and the smell types. Our study focused on Java and Python smells and this choice can limit the generalizability of our results to

other programming languages. Nonetheless, in contrast to existing literature focusing primarily on Java smells, we included Python for cross language smell detection. Moreover, we selected the most commonly investigated smells in both languages from different levels of granularity (class-level and method-level). However, we can't generalize our findings to other smell types (*e.g.* Middle Man smell). Future research should include additional languages and smell types to validate the broader applicability of our approach in cross language smell detection.

We used statistical tests to compare between the models detection performance and whether the observed difference is statistically significant or not. The choice of the employed statistical test as well as the number of runs in our study can influence the conclusion validity of our study results. To mitigate this threat, we employed a nonparametric Wilcoxon signed-rank test to perform the models pairwise comparisons due to data non-normal distribution. In addition, we adjusted the $p$-value by apply Bonferroni correction to reduce the hypothesis testing Type I error. Lastly, the models detection performance was the average of 100 runs, generated by a 10 stratified folds cross validation repeated 10 times. Thus, ensuring unbiased model estimates with low variance.

## RESEARCH IMPLICATIONS

The findings of our conducted empirical study have several important research and practical implications for both software engineers and machine learning practitioners. We aimed in our study to contribute to the knowledge advancement in the field of code smell detection and provides practical insights that can enhance the overall software quality. We can list our study implications as follows:

- Enhanced detection tools. Dynamic stacking ensembles offers a reliable code smell detection models across Java and Python programming languages. These ensemble models can be integrated with code analysis tools to enhance the overall code smell detection capabilities. This improvement can assist software engineers in the refactoring process to counter these smells and enhance the overall software maintenance and quality.
- Practical usability. Dynamic stacking ensembles result in less complex models in comparison to full stacking ensembles without any noticeable performance degrade. This reduction in complexity is achieved through the selection of the most significant base models using GS and BE approaches, reducing both the number of base models and computational resources required. Practitioners can utilize these advanced stacking models in code smell detection with fewer resources. Moreover, our open source code allows practitioners to deploy these models for the purpose of code smell detection in their projects and suggest any further improvements to these ensembles.
- Cross-language applicability. By extending code smell detection to include Python, in addition to the most investigated Java language, our research promotes the development of cross-language code smell detection models that is applicable across multiple languages. Future research should include more programming languages to enhance the generalizability of code smell detection models.

## CONCLUSION

In this article, we introduced two strategies to build dynamic ensembles: greedy search and backward elimination. For the greedy search strategy, we built our dynamic ensemble incrementally in a forward search fashion by adding the most contributing model to the stacking ensemble base model list in each iteration. Conversely, in the backward elimination strategy, we started with a list of candidate base models, and iteratively eliminated the least contributing model from the base models list. We performed a comprehensive empirical study to investigate the effectiveness and stability of dynamic ensembles in detecting four Java and two Python code smells.

Our article presents interesting empirical evidence for machine learning and software engineers regarding the detection capabilities of dynamic stacking ensembles to detect code smells across languages. Our findings can be summarized as follows: (1) Greedy search and backward elimination strategies resulted in different base models and model complexities when constructed to detect different code smells. (2) The backward elimination strategy yielded less complex dynamic ensembles. (3) Dynamic ensembles demonstrated stable and high detection performance across Java and Python code smells. (4) Dynamic ensembles were less complex than full stacking ensembles without any significant performance detection loss.

This work can be extended further into many future directions: First, our empirical study can be replicated with additional Java and Python code smells to examine the stability of dynamic stacking ensembles in detecting these smells. Second, other programming languages (*e.g.*, C# and C++) smells detection can be further investigated using dynamic ensembles to increase the study outcomes generalizability. Third, future work can explore additional strategies (*e.g.* evolutionary algorithms) to provide a more comprehensive evaluation against our investigated strategies. Lastly, we can consider a larger set of candidate base models to build the dynamic ensembles. We selected six versatile models as base models, however a study can be designed to examine the effectiveness of greedy search and backward elimination strategies in building less complex models, when a larger candidate base models is used.

### Funding

This work was supported by King Fahd University of Petroleum and Minerals (KFUPM). The funders had no role in study design, data collection and analysis, decision to publish, or preparation of the manuscript.

### Grant Disclosures

The following grant information was disclosed by the authors:
King Fahd University of Petroleum and Minerals (KFUPM).

### Competing Interests

The authors declare that they have no competing interests.

## Author Contributions

- Hamoud Aljamaan conceived and designed the experiments, performed the experiments, analyzed the data, performed the computation work, prepared figures and/or tables, authored or reviewed drafts of the article, and approved the final draft.

## Data Availability

Code is available at Zenodo:

Aljamaan, H. (2024). Dynamic Stacking Ensemble for Cross-language Code Smell Detection. Zenodo. https://doi.org/10.5281/zenodo.11488752.

The Java datasets are available at ESSeRE Lab: https://essere.disco.unimib.it/machine-learning-for-code-smell-detection/.

The Python datasets are available at Zenodo:

Sandouka, R., & Aljamaan, H. (2023). Python code smells detection using conventional machine learning models. Peerj Computer Science, 9, e1370. https://doi.org/10.5281/zenodo.7512516.

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
