# Peer review of "Dynamic stacking ensemble for cross-language code smell detection"

_PeerJ Computer Science, doi:10.7717/peerj-cs.2254_

## Round 0.1 · original submission · Major Revisions

Based on the reviewers’ comments, you may resubmit the revised manuscript for further consideration. Please consider the reviewers’ comments carefully and submit a list of responses to the comments along with the revised manuscript.

**Language Note:** The review process has identified that the English language must be improved. PeerJ can provide language editing services - please contact us at [email protected] for pricing (be sure to provide your manuscript number and title). Alternatively, you should make your own arrangements to improve the language quality and provide details in your response letter. – PeerJ Staff

Reviewer 1 ·

Basic reporting

The article is well written for Professional English Language viewpoint.
The author has included sufficient literature references. In Literature review section, it would be better to highlight the deficiencies of the current-state-of-the-art.

From Professional article structure perspective, research contributions should be listed in introduction section. Furthermore, a research methodology section should be added, highlighting the research objectives, questions as well as their rationale. Moreover, Threats to validity section needs to be added. The research implication section is also missing.

The formal results are presented. However, the results should be discussed according to formulated research questions.

The references of the considered formulas (equations) need to be provided.

Experimental design

The authors have utilised greedy search and backward
elimination techniques. However, Greedy technique never ensure global optimal solution. No back tracking and also suffer from combinatorial explosion problem. On the other hand, Backward elimination is computationally expensive, and having a risk of over fitting. The author needs to provide the rationale and justification about the selection of considered methodologies.

The rationale of the focused code smells need to be explicitly mentioned.

Validity of the findings

Better to include threats to validity section.

It is suggested to provide the employed data for the study replication purpose.

Cite this review as

Reviewer 2 ·

Basic reporting

Paper provides sufficient background/context for the study. However there is need to include recent related references to enrich it further. Please see detailed report attached herewith.

Experimental design

The study is interesting and these is need of cross language code smell detection studies. Research questions are well defined and relevant. Investigation and experiment is through. There is scope of improvement. Please see report attached herewith.

Validity of the findings

This part is well addressed however it could be elaborated further to strengthen it further,

Annotated reviews are not available for download in order to protect the identity of reviewers who chose to remain anonymous.
Cite this review as

·

Basic reporting

The paper proposes using dynamic ensembles implemented through two strategies: greedy search and backward elimination. The authors argue that these strategies can accurately detect code smells in two programming languages (i.e., Java and Python) and are less complex than complete stacking ensembles.

The author should have explained the criteria for selecting the studies in the literature review section.

The author should have explained the criteria for selecting the studies in the literature review section.

The text needs to present the references appropriately formatted throughout the sections. The Introduction section's first sentence illustrates this: “Code smells refer to poor design and implementation choices by software engineers that compromise overall software quality Yamashita and Moonen (2013)”.

The last paragraph of the Introduction presents the structure of the paper. However, it does not mention the name of each section, which can cause an inevitable confusion for the reader.

Experimental design

The text does not present the criteria for selecting Greedy Search (GS) and Backward Elimination (BE) strategies to select models in the dynamic stacking ensembles. Are there any reasons for this choice? Why not evaluate other strategies?

Validity of the findings

There is no discussion about the threats to the validity of the empirical study. For example, the text does not appropriately discuss the impact of the decision to use code smells from two different programming languages: Java and Python.

Additional comments

The main strength of the paper is the literature review that presents an overview of the use of ensemble learning for code smell detection and the evaluation of an approach based on dynamic stacking ensembles to surpass the effectiveness of complete stacking ensembles to detect code smells.

Cite this review as

---

## Round 0.2 · Minor Revisions

Some of the comments of one of reviewers re-work. Please address these comments and re-submit for further consideration.

Reviewer 2 ·

Basic reporting

Reported in the first review report.

Experimental design

Reported in the first review report.

Validity of the findings

Reported in the first review report.

Additional comments

The author has addressed a few of the comments. However, there is still a need for further revisions to enhance the manuscript's credibility and bring it up to the standard of the journal.

1. It would be beneficial to include specific references that support the descriptions of Greedy Search, Backward Elimination, and other algorithms mentioned in the paper. This will not only enhance the credibility of your research but also provide a solid foundation for further research in the field.

2. In Tables 2 and 3, the step numbering and its corresponding descriptions are not properly aligned, making it difficult to read. It is suggested that the Stacking Ensemble Algorithm and Greedy Search Algorithm be presented in a different format, such as a flowchart or a step-by-step guide, to improve readability. This could involve aligning the steps and descriptions in a more structured manner.

3. It is crucial to provide a detailed explanation of the working principles of the Stacking Ensemble Algorithm, Greedy Search Algorithm, and Backward Elimination Algorithm. Additionally, as complexity is a key factor in algorithm performance evaluation, it is important to mention the time complexities of the proposed algorithm. In the RESEARCH IMPLICATIONS section, you mentioned that 'Dynamic stacking ensembles result in less complex models…', but the complexity of this algorithm is not mentioned in the manuscript. This information is crucial for comprehensively understanding your research and its implications.

4. A few relevant recent references could be included in this paper to enrich the manuscript. A few references belong to code smells utilizing machine learning, and ensemble learnings are not included, as suggested previously.

Cite this review as

·

Basic reporting

No comment. The author addressed all issues reported in the previous review.

Experimental design

No comment. The author addressed all issues reported in the previous review.

Validity of the findings

No comment. The author addressed all issues reported in the previous review.

Additional comments

The author addressed all issues reported in the previous review.

Cite this review as

---

## Round 0.3 · accepted · Accept

Congratulations, the reviewers are satisfied with the revised version of the manuscript and recommended an accept decision.

Reviewer 2 ·

Basic reporting

The paper is well written and organized in logical manner. Background literature is well cited. Formal results include clear definitions and well supported by analysis.

Experimental design

It is now well addressed along with thorough investigation, and analysis.

Validity of the findings

Experiments are well conducted and and conclusions are well articulated.
Results are well supported and meet required standards.

Additional comments

Authors have addressed all comments in satisfactorily manner.
Therefore paper can be accepted for publications.

Cite this review as